# NEURAL MANIFOLD CLUSTERING AND EMBEDDING

## ABSTRACT

Given a union of non-linear manifolds, non-linear subspace clustering or manifold clustering aims to cluster data points based on manifold structures and also learn to parameterize each manifold as a linear subspace in a feature space. Deep neural networks have the potential to achieve this goal under highly non-linear settings given their large capacity and flexibility. We argue that achieving manifold clustering with neural networks requires two essential ingredients: a domain-specific constraint that ensures the identification of the manifolds, and a learning algorithm for embedding each manifold to a linear subspace in the feature space. This work shows that many constraints can be implemented by data augmentation. For subspace feature learning, Maximum Coding Rate Reduction ($MCR^2$) objective can be used. Putting them together yields *Neural Manifold Clustering and Embedding* (NMCE), a novel method for general purpose manifold clustering, which significantly outperforms autoencoder-based deep subspace clustering and achieve state-of-the-art performance on several important benchmarks. Further, on more challenging natural image datasets, NMCE can also outperform other algorithms specifically designed for clustering. Qualitatively, we demonstrate that NMCE learns a meaningful and interpretable feature space. As the formulation of NMCE is closely related to several important Self-supervised learning (SSL) methods, we believe this work can help us build a deeper understanding on SSL representation learning.

## 1 INTRODUCTION

Here we investigate unsupervised representation learning, which aims to learn structures (features) from data without the use of any label. If the data lie in a linear subspace, the linear subspace can be extracted by Principal Component Analysis (PCA) (Jolliffe, 1986), one of the most basic forms of unsupervised learning. When the data occupy a union of several linear subspaces, subspace clustering (SC) (Vidal et al., 2016) is needed to cluster each data point to a subspace as well as estimating the parameters of each subspace. Here we are concerned with even more challenging scenarios, when data points come from a union of several non-linear low-dimensional manifolds. In such scenarios, the clustering problem can be formulated as follows (Elhamifar & Vidal, 2011):

**Task 1.** *Manifold Clustering and Embedding: Given that the data points come from a union of low-dimensional manifolds, we shall segment the data points based on their corresponding manifolds, and obtain a low-dimensional embedding for each manifold.*

Various methods have been developed to solve this problem (Abdolali & Gillis, 2021), but it is still an open question how to use neural networks effectively in manifold clustering problems (Haeffele et al., 2020). In this paper, we propose *neural manifold clustering and embedding* (NMCE) that follows three principles: 1) The clustering and representation should respect a domain-specific constraint, e.g. local neighborhoods, local linear interpolation or data augmentation invariances. 2) The embedding of a particular manifold shall not collapse. 3) The embedding of identified manifolds shall be linearized and separated, i.e. they occupy different linear subspaces. We achieve 1) using data augmentations, and achieve 2) and 3) with the subspace feature learning algorithm maximal coding rate reduction ($MCR^2$) objective (Yu et al., 2020). This work make the following specific contributions:

1. We combine data augmentation with $MCR^2$ to yield a novel algorithm for general purpose manifold clustering and embedding (NMCE). We also discuss connections between the algorithm and self-supervised contrastive learning.

2. We demonstrate that NMCE achieves state-of-the-art performance on standard subspace clustering benchmarks, and can outperform the best clustering algorithms on more challenging high-dimensional image datasets like CIFAR-10 and CIFAR-20. Further, empirical evaluation suggests that our algorithm also learns a meaningful feature space.

## 2 RELATED WORK

**Manifold Learning.** In classical manifold learning, the goal is to map the manifold-structured data points to a low-dimensional representation space such that the manifold structure is preserved. There are two key ingredients: 1) Choosing a geometric property from the original data space to be preserved. For example, the local euclidean neighborhood (Belkin & Niyogi, 2003), or linear interpolation by neighboring data points (Roweis & Saul, 2000). 2) The embedding should not collapse to a trivial solution. To avoid the trivial solution, the variance of the embedding space is typically constrained in spectral-based manifold learning methods.

**Manifold Clustering and Embedding.** When the data should be modeled as a union of several manifolds, manifold clustering is needed in addition to manifold learning. When these manifolds are linear, subspace clustering algorithms (Ma et al., 2007; Elhamifar & Vidal, 2013; Vidal et al., 2016) can be used. When they are non-linear, manifold clustering and embedding methods were proposed. They generally divide into 3 categories (Abdolali & Gillis, 2021): 1. Locality preserving. 2. Kernel based. 3. Neural Network based. Locality preserving techniques implicitly make the assumption that the manifolds are smooth, and are sampled densely (Souvenir & Pless, 2005; Elhamifar & Vidal, 2011; Chen et al., 2018). Additionally, smoothness assumption can be employed (Gong et al., 2012). Our method generalizes those techniques by realizing them with geometrical constraints. The success of kernel based techniques depends strongly on the suitability of the underlying kernel, and generally requires a representation of the data in a space with higher dimension than the data space (Patel & Vidal, 2014). Deep subspace clustering methods, such as Ji et al. (2017); Zhang et al. (2019); Zhou et al. (2018) jointly perform linear subspace clustering and representation of the data, and has the potential to handle high dimensional data effectively. However, it has been shown that most performance gains obtained by those methods should be attributed to an ad-hoc post-processing step applied to the self-expression matrix. Using neural networks only provide a very marginal gain compared to clustering the raw data directly using linear SC (Haeffele et al., 2020). Our work differs from those techniques mainly in two aspects: i) While most of the previous methods were generative (autoencoders, GANs), our loss function is defined in the latent embedding space and is best understood as a contrastive method. ii) While previous methods use self-expression based SC to guide feature learning, ours uses $MCR^2$ to learn the subspace features. Recently, some deep SC techniques also applied data augmentation (Sun et al., 2019; Abavisani et al., 2020). However, in those works, data augmentation played a complementary role of improving the performance. In our method, data augmentation plays a central role for enabling the identification of the clusters.

**Self-Supervised Representation Learning.** Recently, self-supervised representation learning achieved tremendous success with deep neural networks. Similar to manifold clustering and embedding, there are also two essential ingredients: 1) Data augmentations are used to define the domain-specific invariance. 2) The latent representation should not collapse. The second requirement can be achieved either by contrastive learning (Chen et al., 2020), momentum encoder (He et al., 2020; Grill et al., 2020) or siamese network structure (Chen & He, 2021). More directly related to our work is Tao et al. (2021), which proposed feature orthogonalization and decorrelation, alongside contrastive learning. Recently, variance regularization along was also successfully used to achieve principle 2) (Zbontar et al., 2021; Bardes et al., 2021), attaining state-of-the-art SSL representation performance. Part of our method, the total coding rate (TCR) objective achieves a similar effect, see discussion in Appendix A.6. However, beyond self-supervised features, our algorithm additionally show strong clustering performance, and directly learns a meaningful latent space. The simultaneous manifold clustering and embedding in NMCE is also related to online deep clustering (Caron et al., 2018; 2020) method. Also notable is Deshmukh et al. (2021), where the concept of population consistency is closely related to the constraint functional we discussed.

**Clustering with Data Augmentation** Our method use data augmentation to ensure correct clustering of the training data. Although not explicitly pointed out, this is also the case for other clustering techniques (Shen et al., 2021; Tsai et al., 2020; Li et al., 2021). Our understanding of data augmentation is also consistent to works that specifically study the success of data augmentations (HaoChen et al., 2021; von Kügelgen et al., 2021).

## 3 NEURAL MANIFOLD CLUSTERING AND EMBEDDING

### 3.1 PROBLEM SETUP

Let's assume data points $x_i \in \mathbb{R}^d$ are sampled from a union $\bigcup_{j=1}^n X_j$ of manifolds $X_1, X_2, ..., X_n$.[1] As stated in Task 1, the goal of manifold clustering and embedding is to assign each data point to the corresponding manifold (clustering), as well as learning a coordinate for each manifold (manifold learning). To achieve this goal, we use neural network $f$, which learns to map a data point $x$ to the feature embedding $z \in \mathbb{R}^{d_{emb}}$ and the cluster assignment $c \in [1, n]$, i.e. $z, c = f(x)$. The cluster assignment shall be equal to the ground-truth manifold assignment,[2] $z$ should parameterize the coordinates of the corresponding manifold $X_c$.

To make the feature space easier to work with, one can enforce the additional separability requirement that for any $X_j, X_k$ and $j \neq k$, the feature vectors are perpendicular, $Z_j \perp Z_k$. Here $Z_j$ denotes the embedding feature vectors of data points in $X_j$, and we define perpendicular between two sets in the following fashion: If $\tilde{Z}_j \subseteq Z_j, \tilde{Z}_k \subseteq Z_k$ such that $\forall z_j \in \tilde{Z}_j, z_k \in \tilde{Z}_k$, we have $z_j \cdot z_k \neq 0$, then either $\tilde{Z}_j$ or $\tilde{Z}_k$ has zero measure.

In the following, we first argue that to make clustering possible with a neural network, one should define the additional geometric constraint that makes the manifold clusters identifiable. Second, we discuss how to implement the required geometric constraints and combine them with a recently proposed joint clustering and subspace learning algorithm MCR$^2$ (Yu et al., 2020) to achieve neural manifold clustering and embeddng (NMCE).

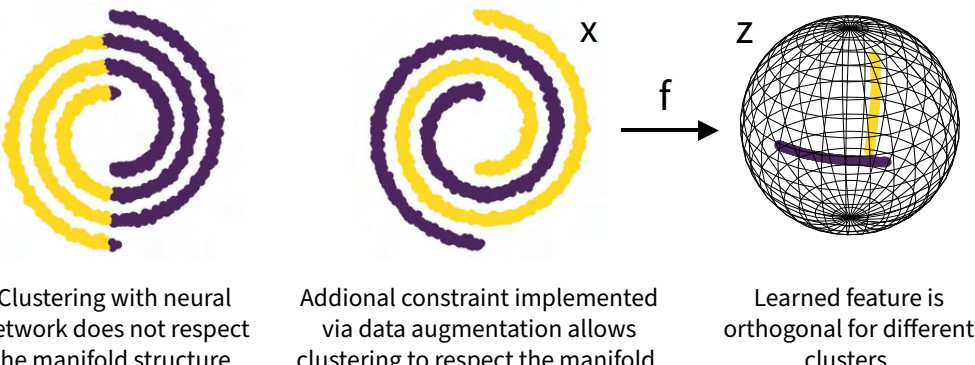

| Clustering with neural network does not respect the manifold structure. | Addional constraint implemented via data augmentation allows clustering to respect the manifold. | Learned feature is orthogonal for different clusters. |

Figure 1: Adding small Gaussian noise data augmentation and requiring the augmented samples of the same point to be assigned to the same cluster and have similar embedding allows neural network to find the desired manifolds.

### 3.2 CLUSTERING ALWAYS INVOLVES IMPLICIT ASSUMPTIONS

Even the simplest clustering algorithms rely on implicit assumptions. For example, in k-means clustering, the implicit assumption is that the clusters in the original data space are continuous in terms of L2 distance. For Linear SC, the assumption is that data points are co-linear within each cluster. If a neural network is trained on example data to learn a cluster assignment function

---

[1]We do not consider topological issues here and assume that all of them are homeomorphic to $\mathbb{R}^{d_i}$ for some $d_i$

[2]Up to a permutation since the training is unsupervised.

$c = f(x)$, the resulting clustering will be arbitrary and not resemble the solution of k-means or linear SC clustering. This is because neural networks are flexible and no constraint is imposed on the clustering function to force the result to respect the geometry of the original data. One example of this is shown in Figure 1 left panel, where a deep clustering method outputs a rather arbitrary clustering for the double spiral data.

To make the clusters learnable for a neural network, one needs to introduce constraints explicitly. In the example in Figure 1, one can easily reason that, to be able to separate the two spirals from each other, the clustering function needs to ensure that all close neighbors of a point from the data distribution are assigned to the same cluster, essentially the same assumption implicitly made in locality-based manifold clustering algorithms (Abdolali & Gillis, 2021). We formalize the notion of constraints to a constraint functional $D(f)$ (for a specific data distribution) that has the following property: All cluster assignment functions $f$ that makes $D$ attain its minimum value (assumed to be 0) and also cluster data points to the ground truth number of clusters will correctly cluster all data points. For example, we can construct a functional $D$ that takes value $D = 0$ for all clustering functions that satisfy the locality constraint on the dataset, and $D > 0$ otherwise. This notion of constraint function is completely general. For example, linear subspace clustering is recovered if the constraint function take value 0 if and only if the found clusters are linear subspace.

In practice, one usually cannot optimize the neural network clustering function $f$ subject to $D = 0$, since the correct solution would have to be found at initialization. A more practical way to use the constraint function is to use the relaxed objective with weighting $\lambda$:

$$L(f) = L_{clst}(f) + \lambda * D(f) \tag{1}$$

where $L_{clst}$ is some objective function that will force $f$ to cluster the dataset. With a suitable $\lambda$, optimizing this objective leads to learning of the correct clusters, by the assumption above. To achieve manifold clustering, one just needs to find the appropriate constraint functional.

### 3.3 SUBSPACE FEATURE LEARNING WITH MAXIMUM CODING RATE REDUCTION

Having introduced explicit constraints for learning the manifold clustering with neural networks, we still need a concrete algorithm for learning a linear subspace-structured representation given the clustering (manifold learning). Fortunately, the recently proposed principle of Maximum Coding Rate Reduction ($\text{MCR}^2$) (Yu et al., 2020) provides a principled learning objective for this purpose. We denote the dataset (union of all manifolds) by $\mathsf{X}$, and random variable that represent distribution on each manifold $\mathsf{X}_i$ by $\mathcal{X}_i$. For a certain encoder $\mathcal{Z} = f(\mathcal{X})$ (without clustering output), $z \in \mathbb{R}^{d_{emb}}$, the Gaussian coding rate function is defined to be:

$$R(\mathcal{Z}, \epsilon) = \frac{1}{2}\text{logdet}(\mathbf{I} + \frac{d_{emb}}{\epsilon^2}\text{cov}(\mathcal{Z})) \tag{2}$$

where cov denotes the covariance matrix function for a vector random variable: $\text{cov}(\mathcal{Z}) = \mathbf{E}_{p(z)}[zz^T]$. This function is approximately the Shannon coding rate of a multivariate Gaussian distribution given average distortion $\epsilon$ (Cover, 1999), and can be motivated by a ball-packing argument. However, we focus on its geometric implication and do not discuss the information-theoretic aspect further. Readers interested to the full introduction of this objective can refer to the original papers (Yu et al., 2020; Ma et al., 2007).

Suppose for now that we are also given the cluster assignment function $c(x)$ that outputs the manifold index $c(x) = i$ for $x \in \mathsf{X}_i$. We can then calculate the average coding rate for a particular cluster $i$ as $R(\mathcal{Z}_i, \epsilon)$, where $\mathcal{Z}_i = f(\mathcal{X}_i)$. The $\text{MCR}^2$ principle states that, to learn a subspace-structured representation, one needs to optimize $f$ by maximizing the difference between the coding rate of all of $\mathcal{Z}$ and the sum of coding rate for each cluster $\mathcal{Z}_i$:

$$\Delta R(\mathcal{Z}, c, \epsilon) = R(\mathcal{Z}, \epsilon) - \sum_{i=1}^{n} R(\mathcal{Z}_i, \epsilon) \tag{3}$$

It has been shown that $\text{MCR}^2$ guarantees the perpendicularity requirement in the problem setup above. Theorem A.6 in Yu et al. (2020) states that under the assumption that $\text{rank}(\mathsf{Z}_i)$ is bounded, and the coding error $\epsilon$ is sufficiently low, maximizing the $\text{MCR}^2$ objective guarantees that $\mathsf{Z}_i \perp \mathsf{Z}_j$ for any $i \neq j$, and the rank of each $\mathsf{Z}_i$ is as large as possible. For full details and proof, see Yu et al. (2020).

### 3.4 Neural Manifold Clustering and Embedding

The MCR$^2$ principle can be extended to the case where the labeling function $c(x)$ is not given, but is instead learned jointly by optimizing the MCR$^2$ objective. In this case, we fuse the clustering function into $f$: $z, c = f(x)$, note that now $\mathcal{Z}_i$ depends on $f$ implicitly. However, as discussed earlier, one will not be able to find the correct clusters by using MCR$^2$ alone, as it is unconstrained in the data space. An additional constraint term $D(f)$ has to be included to make the clusters identifiable. This leads to the Neural Manifold Clustering and Embedding (NMCE) objective:

$$L_{NMCE}(f, \epsilon) = \sum_{i=1}^{n} R(\mathcal{Z}_i, \epsilon) - R(\mathcal{Z}, \epsilon) + \lambda D(f) \tag{4}$$

It is possible to replace the MCR$^2$ objective in (4) by any other objective for subspace feature learning, however, this generalization is left for future work. Given a suitable constraint functional $D$, the NMCE objective will guide the neural network to cluster the manifold correctly and also learn the subspace-structured feature, thus achieving manifold clustering and embedding. We explain in the next section how $D$ can be practically implemented.

### 3.5 Implementing constraints via data augmentation

Here we propose a simple method to implement various very useful constraints. The proposed method uses data augmentations, a very common technique in machine learning, and enforce two conditions on the clustering and embedding function $f$: 1. Augmented data points generated from the same point should belong to the same cluster. 2. The learnt feature embedding of augmented data points should be similar if they are generated from the same point. We use $\mathcal{T}(x)$ to represent a random augmentation of $x$: $c, z = f(\mathcal{T}(x))$, $c', z' = f(\mathcal{T}'(x))$, then $D(f) = E_{p(x)}\mathsf{sim}(z, z')$. Here, $\mathsf{sim}$ is a function that measures similarity between latent representations, for example cosine similarity, or L2 distance. The requirements $c = c'$ can be enforced by using average of $c$ and $c'$ to assign $z$ and $z'$ to clusters during training.

For the double spiral example, the augmentation is simply to add a small amount of Gaussian noise to data points. This will have the effect of forcing neighboring points in the data manifold to also be neighbors in the feature space and be assigned to the same cluster. The result of using this constraint in the NMCE objective can be seen in Figure 1 middle and right panel. For more difficult datasets like images, one needs to add augmentations to constrain the clustering in the desired way. The quality of clustering is typically measured by how well it correspond to the underlying content information (object class). Therefore, we need augmentations that perturbs style information (color, textures and background) but preserves content information, so that the clustering will be based on content but not style. Fortunately, extensive research in self-supervised contrastive learning has empirically found augmentations that achieves that very effectively (Chen et al., 2020; von Kügelgen et al., 2021).

### 3.6 Multistage training and relationship to self-supervised contrastive learning

In practice, we find that for all but the simplest toy experiments it is difficult to optimize the full NMCE objective (Eq. 4) from scratch. This is because the clusters are incorrectly assigned at the start of training, and the sum in Eq 4 effectively cancels out the total coding rate term $R(\mathcal{Z}, \epsilon)$. To achieve good performance, it is critical to achieve high total coding rate, and high similarity between augmented samples in the feature space. We thus resort to a multistage training procedure, with the first stage always being optimizing the following objective:

$$L_{TCR}(f, \epsilon) = -R(\mathcal{Z}, \epsilon) + \lambda D(f) \tag{5}$$

We call this the Total Coding Rate (TCR) objective, which is a novel self-supervised learning objective by itself. Through simple arguments (Appendix A.6) one can see that this objective encourages the covariance matrix of $\mathcal{Z}$ to be diagonal. Along with a similarity constraint between augmented samples, this objective turns out to asymptotically achieve the same goal as VICReg (Bardes et al., 2021) and BarlowTwins (Zbontar et al., 2021). We discuss this further in Appendix A.6.

After training with the TCR objective, we found that usually the feature $\mathcal{Z}$ already possess approximate subspace structure. For simple tasks, the features can be directly clustered with standard linear SC techniques such as EnSC (You et al., 2016). For more difficult tasks, we found that the full NMCE objective performs much better.

## 4 RESULTS

We provide code to reproduce the double spiral toy example in the supplementary material, the full code will be released upon publication.

For the synthetic experiments, a MLP backbone is used as backbone and two linear last layers are used to produce feature and cluster logits output, ELU is used as the activation function. For all image datasets, standard ResNet with ReLU activation and various sizes is used as the backbone. After the global average pooling layer, one linear layer with 4096 output units, batch normalization and ReLU activation is used. After this layer, two linear heads are used to produce feature and cluster logits outputs. The number of clusters used is always equal to the ground truth. For all experiments, two augmented samples, or "views", are used for each data sample in a batch. Gumbel-Softmax (Jang et al., 2016) is used when the cluster assignment is learned. Similar to Yu et al. (2020), feature vectors are always normalized to the unit sphere. The constraint term $D$ is always cosine similarity between two augmented samples. Cluster assignment probabilities and feature vectors are averaged between augmentations before sending into the NMCE loss. The regularization strength $\lambda$ is always determined empirically. Hyper-parameters and further details are available in the Appendix B.

Table 1: Clustering performance comparison on COIL20 and COIL100. Listed are error rates, NMCE is our method, see text for references for other methods.

| Dataset | SSC | KSSC | AE+EDSC | DSC | S2ConvSCN | MLRDSC-DA | NMCE (Ours) |
|---------|-----|------|---------|-----|-----------|-----------|-------------|
| COIL20  | 14.83 | 24.65 | 14.79 | 5.42 | 2.14 | 1.79 | **0.0** |
| COIL100 | 44.90 | 47.18 | 38.88 | 30.96 | 26.67 | 20.67 | **11.53** |

### 4.1 SYNTHETIC AND IMAGE DATASETS

For the toy example in Figure 1, data augmentation is simply adding a small amount of noise, and full NMCE objective is directly used, the clustering output is jointly learned.

To verify the NMCE objective, we perform synthetic experiment by clustering a mixture of manifold-structured data generated by passing Gaussian noise through two randomly initialized MLPs. Small Gaussian noise data augmentation is used to enforce the locality constraint. A two stage training process is used, the first stage uses TCR objective while the second stage uses full NMCE objective. Much like the toy experiment, the result is consistent with our interpretation, see Appendix A.1 for results and details.

To compare NMCE with other subspace clustering methods, we apply it to COIL20 (Nene et al., 1996a) and COIL100 (Nene et al., 1996b), both are standard datasets commonly used in SC literature. They consist of images of objects taken on a rotating stage in 5 degree intervals. COIL20 has 20 objects with total of 1440 images, and COIL100 has 100 objects with total of 7200 images. Images are gray-scaled and down-sampled by 2x. For COIL20, a 18-layer ResNet with 32 filters are used as the backbone, the feature dimension is 40. For COIL100, and 10-layer ResNet with 32 filters is used, and feature dimension is 200. We determined the best augmentation policy for this experiment by manual experimentation on COIL20, and the same policy is applied to COIL100. Details on the searched policy can be found in Appendix B. For this experiment, we use EnSC (You et al., 2016) to cluster features learned with TCR objective. We found this procedure already performs strongly, and full NMCE objective is not necessary.

We compare with classical baseline such as Sparse Subspace Clustering (SSC) (Elhamifar & Vidal, 2013) as well as recent deep SC techniques including KSSC(Patel & Vidal, 2014), AE+EDSC(Ji et al., 2014), DSC(Ji et al., 2017), S2ConvSCN(Zhang et al., 2019) and MLRDSC-DA(Abavisani

et al., 2020). As can be seen in Table 1, our method achieves a perfect result of 0.0 error rate for COIL20, and error rate of 11.53 for COIL100, substantially outperforming previous state of the art of 1.79 and 20.67 error rate. This indicates that NMCE can truly leverage the non-linear processing capability of deep networks, unlike previous deep SC methods, which only marginally outperform linear SC on the pixel space (Haeffele et al., 2020).

## 4.2 SELF-SUPERVISED LEARNING AND CLUSTERING OF NATURAL IMAGES

Recently, unsupervised clustering has been extended to more challenging image datasets such as CIFAR-10, CIFAR-20 (Krizhevsky et al., 2009) and STL-10 (Coates et al., 2011). The original $MCR^2$ paper (Yu et al., 2020) also performed those experiments, but they used augmentation in a very different way (See Appendix A.2 for a discussion). The CIFAR-10 experiment used 50000 images from the training set, CIFAR-20 used 50000 images from training set of CIFAR-100 with 20 coarse labels. The STL-10 experiment used 13000 labeled images from original train and test set. ImageNet-10 and ImagetNet-Dogs used 13000 and 19500 images subset from ImageNet, respectively (Li et al., 2021). We use ResNet-18 as the backbone to compare with $MCR^2$, and use ResNet-34 as the backbone to compare with other clustering methods. In both cases, the feature dimension is 128. Standard image augmentations for self-supervised learning is used (Chen et al., 2020).

Table 2: Supervised evaluation performance for different feature types and evaluation algorithms. See text for details.

| Model | Proj | Pre-feature | Pool | Proj (16 avg) | Pre-feature (16 avg) | Pool (16 avg) |
|---|---|---|---|---|---|---|
| SVM | 0.911 | 0.889 | 0.895 | 0.922 | 0.905 | **0.929** |
| kNN | 0.904 | 0.851 | 0.105 | **0.910** | 0.800 | 0.103 |
| NearSub | 0.898 | 0.902 | 0.903 | 0.903 | 0.909 | **0.911** |

### 4.2.1 TRAINING STRATEGY, SUPERVISED EVALUATION

Here we use a three stage training procedure: 1. Train with TCR objective. 2. Reinitialize the last linear projection layer and add a cluster assignment layer. Then, freeze parameters in the backbone network and train the two new layers with full NMCE objective. 3. Fine tune the entire network with NMCE objective.

As discussed earlier, the first stage is very similar to self-supervised contrastive learning. Therefore, we study features learned at this stage with supervised evaluation, the result with ResNet-34 is listed in Table 2. To evaluate the learned features, we use SVM, kNN and a Nearest Subspace classifier (NearSub). We use SVM instead of linear evaluation training because we find that it is more stable and insensitive to particular parameter settings. Here we compare three types of features, Proj: 128d output of the projector head. Pre-feature: 4096d output of the linear layer after backbone. Pool: the 2048d feature from the last average pooling layer. We find that averaging features obtained from images processed by training augmentations improves the result. Therefore we also compare performance in this setting and denote it by 16avg, as 16 augmentations are used per image. Images from both training and test set are augmented. As can be seen from the table, without augmentation, SVM using the projector head feature gives the best accuracy. This is surprising, as most self-supervised learning techniques effectively use Pool features because the performance obtained with Proj features is poor. When averaging is used, a SVM with Pool feature becomes the best performer, however, SVM and kNN with Proj feature also achieve very high accuracy. In Appendix, we further compared Proj and Pool features using SVM in cases where limited labeled data are available. In such cases, Proj feature significantly outperform Pool feature, showing that the TCR objective encourages learning of meaningful feature in the latent space – unlike other algorithms, where the gap between performance of Pool and Proj features can be very large (Chen et al., 2020).

Table 3: Clustering performance comparisons. Clustering: clustering specific methods. SC: subspace clustering methods. See text for details.

| Models | CIFAR-10 | | | CIFAR-20 | | | STL-10 | | | ImageNet-10 | | | ImageNet-Dogs | | |
|---|---|---|---|---|---|---|---|---|---|---|---|---|---|---|---|
| | ACC | NMI | ARI | ACC | NMI | ARI | ACC | NMI | ARI | ACC | NMI | ARI | ACC | NMI | ARI |
| **Clustering** | | | | | | | | | | | | | | | |
| k-means | 0.525 | 0.589 | 0.276 | 0.130 | 0.084 | 0.028 | 0.192 | 0.125 | 0.061 | 0.241 | 0.119 | 0.057 | 0.105 | 0.055 | 0.020 |
| Spectral | 0.455 | 0.574 | 0.256 | 0.136 | 0.090 | 0.022 | 0.159 | 0.098 | 0.048 | 0.271 | 0.151 | 0.076 | 0.111 | 0.038 | 0.013 |
| CC | 0.790 | 0.705 | 0.637 | 0.429 | 0.431 | 0.266 | **0.850** | **0.764** | **0.726** | 0.893 | 0.859 | 0.822 | 0.429 | 0.455 | 0.474 |
| CRLC | 0.799 | 0.679 | 0.634 | 0.425 | 0.416 | 0.263 | 0.818 | 0.729 | 0.682 | - | - | - | - | - | - |
| MoCo Baseline | 0.776 | 0.669 | 0.608 | 0.397 | 0.390 | 0.242 | 0.728 | 0.615 | 0.524 | - | - | - | 0.338 | 0.347 | 0.197 |
| MiCE | 0.835 | 0.737 | 0.698 | 0.440 | 0.436 | 0.280 | 0.752 | 0.635 | 0.575 | - | - | - | 0.439 | 0.428 | 0.286 |
| TCC | **0.906** | 0.790 | 0.733 | 0.491 | 0.479 | 0.312 | 0.814 | 0.732 | 0.689 | 0.897 | 0.848 | 0.825 | 0.546 | 0.512 | 0.409 |
| ConCURL | 0.846 | 0.762 | 0.715 | 0.479 | 0.468 | 0.303 | 0.749 | 0.636 | 0.566 | **0.958** | **0.907** | **0.909** | **0.695** | **0.630** | **0.531** |
| IDFD | 0.815 | 0.711 | 0.663 | 0.425 | 0.426 | 0.264 | 0.756 | 0.643 | 0.575 | 0.954 | 0.898 | 0.901 | 0.591 | 0.546 | 0.413 |
| **SC** | | | | | | | | | | | | | | | |
| $MCR^2$-EnSC | 0.684 | 0.630 | 0.508 | 0.347 | 0.362 | 0.167 | 0.491 | 0.446 | 0.290 | - | - | - | - | - | - |
| $MCR^2$-ESC | 0.653 | 0.629 | 0.438 | - | - | - | - | - | - | - | - | - | - | - | - |
| $MCR^2$-SENet | 0.765 | 0.655 | 0.573 | - | - | - | - | - | - | - | - | - | - | - | - |
| **Ours** | | | | | | | | | | | | | | | |
| NMCE-Res18 | 0.830 | 0.761 | 0.710 | 0.437 | 0.488 | 0.322 | 0.725 | 0.614 | 0.552 | 0.906 | 0.819 | 0.808 | 0.398 | 0.393 | 0.227 |
| NMCE-Res34 | 0.891 | **0.812** | **0.795** | **0.531** | **0.524** | **0.375** | 0.711 | 0.600 | 0.530 | - | - | - | - | - | - |

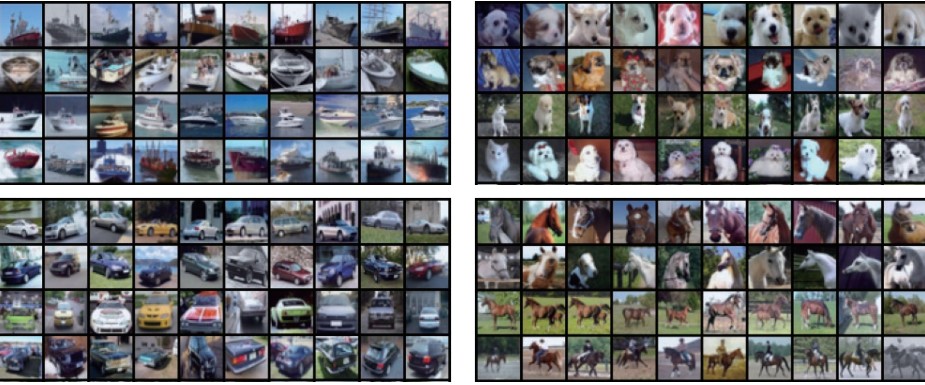

Figure 2: Visualization of principal components from subspace representation learned from CIFAR-10. Rows in each panel are training images that has the largest cosine similarities with different principle components of that subspace. Similarity of images within each component is apparent. For a more complete visualization, see Figure A.6.

### 4.2.2 CLUSTERING PERFORMANCE

We compare clustering performance of NMCE after all 3 stages of training to other techniques. Due to space limits, we only compare with important baseline methods as well as highest performing deep learning based techniques, rather than being fully comprehensive. For a more complete list of methods, see for example Shen et al. (2021). We follow the convention in the field and use clustering Accuracy (ACC), Normalized Mutual Information (NMI), and Adjusted Rand Score (ARI) as metrics. For definitions, see Yu et al. (2020), for example. We list results in Table 3, K-means(MacQueen et al., 1967) and Spectral clustering(Ng et al., 2002) results were adopted from Zhang et al. (2021) and Shen et al. (2021), we always selected the best result. For deep learning based techniques, we compare our ResNet-34 results with CC(Li et al., 2021), CRLC(Do et al., 2021), MiCE(Tsai et al., 2020), TCC(Shen et al., 2021), ConCURL(Deshmukh et al., 2021), and IDFD(Tao et al., 2021). We did not include Park et al. (2021) or Kinakh et al. (2021) since the former is a post processing step, and the later uses a special network architecture. We also separately compare ResNet-18 result with features learned by $MCR^2$ ($MCR^2$-CTRL to be precise, see Appendix A.2) and clustered by EnSC(You et al., 2016), ESC(You et al., 2018) or SENet(Zhang et al., 2021). They are denoted by $MCR^2$-EnSC, $MCR^2$-ESC and $MCR^2$-SENet, respectively. For

MCR$^2$-EnSC, we adopt the result from Yu et al. (2020), since the result is higher and include all 3 datasets, MCR$^2$-ESC and MCR$^2$-SENet results are from Zhang et al. (2021).

Compared with the result from MCR$^2$, our method achieved a substantial gain in all metrics considered. For example, the accuracy is improved by more than 6% on CIFAR-10, and by 9% on CIFAR-20. This shows that we used data augmentation in a much more effective way than original MCR$^2$. Our method can also outperform previous techniques which were specifically optimized for clustering, this happens on most CIFAR-10 and CIFAR-20 metrics, for example.

In the experiments above, stage 1 learns pre-features that are approximately subspace-structured, and stage 2 can be seen as performing subspace clustering on the pre-feature output, as MCR2 was originally a subspace clustering objective (Ma et al., 2007). For CIFAR and STL-10 experiments, fine tuning the entire network with the NMCE objective (stage 3) improves performance by a small but significant amount, see Appendix A.3). Features learned from stage 1 of the training can also be directly clustered by methods like EnSC, but the results were rather poor for the image dataset. It seems that NMCE is a more robust way to cluster the learned features for the image experiments.

After all 3 training stages, one can visualize the structure of each subspace by performing PCA and displaying the training examples that have the highest cosine similarity to each of the principle components. In Figure 2 we show this for 4 out of 10 clusters of CIFAR-10. As can be seen, principle components in each cluster correspond to semantically meaningful sub-clusters. For example, the first row in "dog" cluster are mostly close-ups of white dogs. This shows that after training with the full NMCE objective, samples are clustered based on content similarity in the latent space, instead of being distributed uniformly on the hyper-sphere, as training with just the TCR objective or other self-supervised learning methods would encourage Wang & Isola (2020); Durrant & Leontidis (2021); Zbontar et al. (2021); Bardes et al. (2021); Zimmermann et al. (2021).

## 5 Discussion

In this paper we proposed a general method for manifold clustering and embedding with neural networks. It consists of adding geometric constraint to make the clusters identifiable, then performing clustering and subspace feature learning. In the special case where the constraint is implemented by data-augmentation, we discussed its relationship to self-supervised learning and demonstrated its competitive performance on several important benchmarks.

In the future, it is possible to extend this method beyond data augmentation to other type of constraint functions, or improves its performance using a stronger subspace feature learning algorithm.

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

## A    ADDITIONAL RESULTS AND DISCUSSIONS

### A.1    SYNTHETIC EXPERIMENT

We verify the proposed manifold clustering and embedding algorithm by a simple synthetic experiment. Ground truth data from a union of two manifolds with dimension 3 and 6 is generated by passing 3d and 6d iid. Gaussian noise through two randomly initialized neural networks with Leaky-ReLU activation function (negative part slop 0.2). Data augmentation is Gaussian noise with manitude 0.1. Training used two stages, the first stage used only TCR objective, the second stage with full NMCE objective.

Two situations are examined. One where the random neural network also has randomly initialized biases, which will cause the two manifold to be far apart, making locality constraint implemented by noise augmentation sufficient for identifying the manifolds. Another situation is when bias is not used, the two manifolds then intersect at 0, where the density is rather high. This makes the manifolds not identifiable with only locality constraint.

Results are listed in Table A.1. As can be seen, when the clusters are identifiable, NMCE is able to correctly cluster the data points, as well as learn latent features that is perpendicular between different clusters. At the same time, the feature is not collapsed within each cluster, since if so the average cosine similarity within cluster would be 1. When the clusters are not identifiable, they cannot be learned correctly.

Table A.1: Result from synthetic experiment. Accuracy is in % (chance level is 50%), z-sim is the average absolute value of cosine similarity between feature vectors $z$ for different pairs of $z$. True Cluster: pairs of $z$ are from different ground truth clusters. Found Clusters: pairs of $z$ are from two different found clusters. Within Cluster: pairs of $z$ are randomly picked from the same found cluster, averaged between two found clusters.

| Dataset | Accuracy | z-sim: True Cluster | z-sim: Found Clusters | z-sim: Within Cluster |
|---|---|---|---|---|
| Identifiable | 100.0 | 0.017 | 0.017 | 0.503 |
| Not-identifiable | 69.8 | 0.717 | 0.287 | 0.770 |

### A.2    RELATIONSHIP TO SELF-SUPERVISED LEARNING WITH $MCR^2$

The original paper (Yu et al., 2020) performed self-supervised learning using $MCR^2$ objective without any additional term. Their method treats different augmentations of the same image as a self-labeled subspace. They used large number of augmentations (50) of each image, with only 20 images in each batch. The performance of this method is rather poor, which is expected based on our understanding. In this case, augmented images will form a subspace with certain dimension in the feature space, thus large amount of information about augmentations will be preserved in the latent space. Clustering can then utilize style information and not respect class information.

To improve performance, a variant called $MCR^2$-CTRL is developed, where the total coding rate term is down-scaled. This variant performs significantly better, and is also used in our comparison. This result is also expected, since decreasing the subspace term effectively contract different augmentations of the same image in the feature space, making the feature better respect the constraint needed for correct clustering. However, since the total coding rate is not high in this case, the feature is not diverse enough to achieve good performance.

### A.3    FINE-TUNING BACKBONE WITH NMCE OBJECTIVE

For CIFAR-10, CIFAR-20 and STL-10 experiments, the first training stage already learns very strong self-supervised features, which is then clustered into subspaces in the second stage with backbone network frozen. The clustering performance is already quite good after this stage. In the third stage, the backbone is fine-tuned, which further improves clustering performance. In Table A.2, we show

the effect of fine-tuning backbone network on CIFAR-10 and CIFAR-20 experiment with ResNet-18. As can be seen, fine-tuning produces a small but noticeable gain in clustering performance for all metrics and both datasets. This indicates that using the full NMCE objective can improve performance. If the optimization issue can be resolved, and the entire network is trained with the NMCE objective from scratch, the performance may be further improved.

Table A.2: Fine tuning backbone with NMCE objective. Results shown are from ResNet-18. Fine tuning backbone improves result slightly but notably.

| Model | ACC | NMI | ARI |
|---|---|---|---|
| CIFAR-10 before | 0.819 | 0.743 | 0.690 |
| CIFAR-10 after | **0.830** | **0.761** | **0.710** |
| CIFAR-20 before | 0.422 | 0.471 | 0.300 |
| CIFAR-20 after | **0.437** | **0.488** | **0.322** |

### A.4 COMPARE POOL AND PROJ FEATURE ON LOW DATA CLASSIFICATION

Here we compare SVM accuracy of Pool and Proj features from CIFAR-10 ResNet-18 experiment. The feature averaged over 16 augmentations. The accuracy is plotted as a function of the number of labeled training samples used, see Figure A.1.

As can be seen, Proj feature clearly outperforms Pool feature when few labeled examples are available. This makes Proj feature much more useful than Pool feature, since labeled examples are often scarce real applications.

Note that what shown here is obviously not the optimal way to use labeled example, if one further leverage the clustering information, accuracy should reach 90 with only 10 labeled examples.

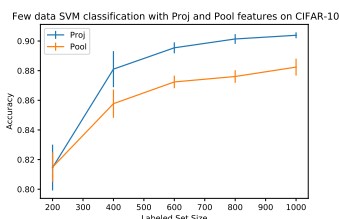

Figure A.1: CIFAR-10 SVM test accuracy plotted against number of labeled examples used for Pool and Proj feature from ResNet-18 experiment. Features averaged over 16 augmentations is used. Error bar is std. over 10 random sampling of training examples.

### A.5 EFFECT OF LAMBDA PARAMETER

Here we study the effect of $\lambda$ parameter that balances the constraint and subspace feature learning term. CIFAR-10 self-supervised accuracy with SVM and kNN evaluation on Proj feature is listed in Table A.3. As can be seen, the accuracy is reasonable in a range of $\lambda$ spanning more than 3x low to high, indicating that the quality of the learned feature is not very sensitive to this parameter.

Table A.3: Effect of parameter $\lambda$.

| $\lambda$ | 20 | 30 | 40 | 50 | 60 | 70 |
|---|---|---|---|---|---|---|
| Proj SVM acc | 0.899 | 0.902 | 0.903 | 0.902 | 0.903 | 0.903 |
| Proj kNN acc | 0.890 | 0.895 | 0.894 | 0.896 | 0.895 | 0.897 |

## A.6 UNDERSTANDING THE FEATURE SPACE

Here we show that the TCR objective used in the first stage in our training procedure for CIFAR and STL-10 datasets theoretically achieve the same result as the recently proposed VICReg (Bardes et al., 2021) and BarlowTwins(Zbontar et al., 2021), both are self-supervised learning algorithms. The optimization target of the two techniques are both making the covariance matrix of latent vector $Z$ approach the diagonal matrix. The first stage training using TCR essentially achieves the same result. To see this, we note the following property of the coding rate function (Kang et al., 2015): For any $Z \in \mathbb{R}^{m \times d}$:

$$\mathsf{logdet}(\mathbf{I} + Z^T Z) = \sum_{i=1}^{\min(m,d)} \log(1 + \sigma_i^2) \tag{6}$$

Where $\sigma_i$ is the $i$th singular value of $Z$. Additionally, we have: $\sum_{i=1}^{\min(m,d)} \sigma_i^2 = ||Z||_{\mathbf{F}}^2$, which follows easily from $||Z||_{\mathbf{F}}^2 = \mathsf{tr}(Z^T Z)$. Since the function $\log(1 + x)$ is concave, the optimization problem $\max_{x_1, x_2, \dots, x_n} \sum_{i=1}^{n} \log(1 + x_i)$ given $\sum_{i=1}^{n} x_i = C$ reaches maxima when all $x$ are equal to each other. Since we normalize the row of $Z$, $||Z||_{\mathbf{F}}^2 = m$, optimization of Equation 6 result in solution with uniform singular value, which is equivalent to diagonal covariance.

We could not successfully reproduce VICReg in our setup due to the large amount of hyper-parameters that needs to be tuned. Therefore we resort to the open-source library solo-learn (da Costa et al., 2021), which provided VICReg implementation. Running the provided script for VICReg produced accuracy of 91.61% on CIFAR-10. We also implemented TCR objective in solo-learn library. Running TCR obtained accuracy of 92.1%. All hyper-parameters and augmentations are the same as VICReg, except batch size is 1024 instead of 256, projection dimension is 64 instead of 2048. Larger batch size or smaller projection dimension didn't work for VICReg, so we stayed with the original parameters.

The covariance matrices of learned feature for SimCLR, VICReg and TCR computed over entire training set are visualized in Figure A.2. For VICReg, first 128 dimension is visualized out of 2048. As can be seen, the diagonal structure is visible in SimCLR feature space, TCR feature space is the closest to diagonal matrix. VICReg feature space is also quite close to diagonal, but the off-diagonal terms seems noisier. Additionally, we plot normalized singular values for VICReg projection space in Figure A.4. This can be compared to TCR and SimCLR singular values in Figure A.3 a). As can be seen, TCR achieves flatter singular value distributions than VICReg, neither SimCLR or VICReg are close to full rank in projection space.

We demonstrate subspace structure of feature space after clustering with full NMCE objective by plotting singular values of each learned subspace in Figure A.3 b). Each subspace found are around rank 10. In all other panels of Figure A.3, we display samples whose feature vector has the highest cosine similarity to the top 10 principle components of each subspace. One can see that most principle components represent a interpretable sub-cluster within each class (even if the sub-cluster is of a different class than the cluster). Same as in Figure 2, feature vectors calculated by averaging 16 augmented images are used.

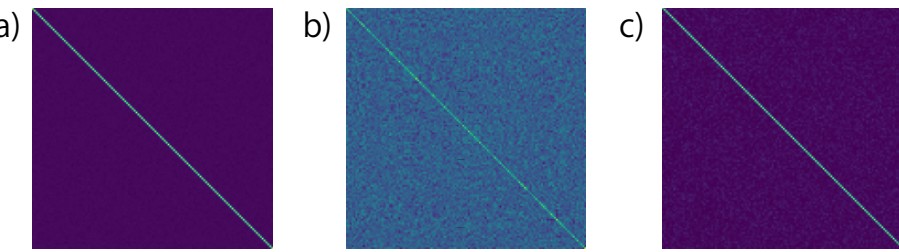

Figure A.2: a), b), c) Covariance matrix of feature vector computed over training set for TCR, SimCLR and VICReg. VICReg result is slightly noisier on the diagonal than TCR. VICReg result is the first 128 dimensions out of 2048, see text for details.

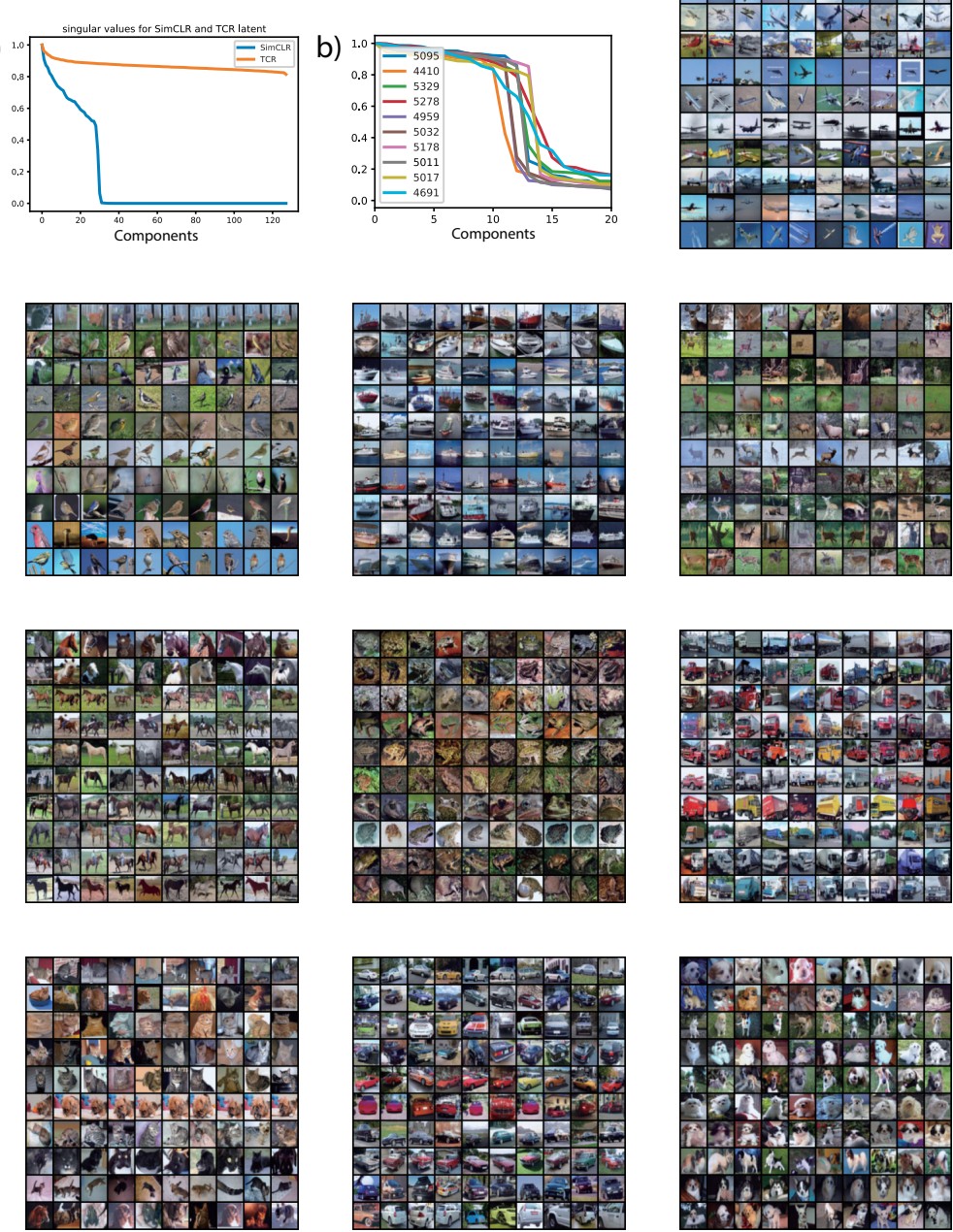

Figure A.3: a) Singular values of feature vector distribution for SimCLR and TCR. b) Singular values for subspaces learned after all 3 training stages. The rest of panels: visualizing 10 training examples most similar to principal components of each clustered subspaces.

# B    EXPERIMENTAL DETAILS

We list hyper-parameters for each experiment in Table B.1.

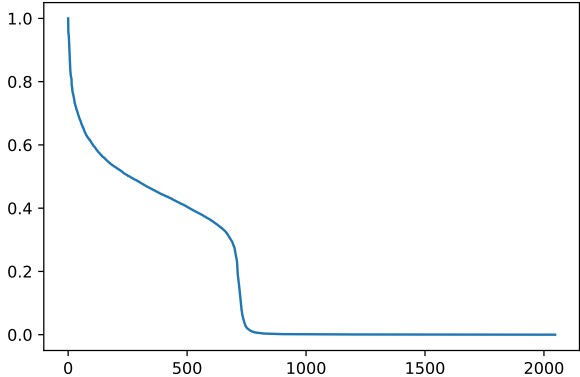

Figure A.4: Singular values of feature vector distribution for VICReg using ResNet-18. VICReg uses 2048d feature space.

### B.1 TOY AND SYNTHETIC EXPERIMENTS

The toy dataset consists of double spiral with radius approximately equal to 15 and Gaussian noise magnitude of 0.05, samples are generated online for each batch. Data augmentation is Gaussian noise with magnitude 0.05.

### B.2 COIL-20 AND COIL-100

The augmentation policy we found with manual search on COIL20 is (all are from torchvision transforms): 1. Random Horizontal Flip with $p = 0.5$. 2. RandomPerspective with magnitude 0.3 and $p = 0.6$ 3. ColorJitter with manitude (0.8, 0.8, 0.8, 0.2), always applied. The entire dataset is passed as a single batch.

### B.3 CIFAR-10, CIFAR-20, STL-10, IMAGENET-10 AND IMAGENET-DOGS

For CIFAR-10 and CIFAR-20, we use standard ResNet-18 and ResNet-34 with 64 input filters. The first layer uses 3x3 kernel, and and no max pooling is used. For other experiments, we use standard ResNet-18 and ResNet-34 with 5x5 first layer kernel and max pooling. For COIL-20 and COIL-100 experiment, 32 input filters is used, and the ResNet-10 is obtained by reducing number of blocks in each stage in ResNet-18 to 1. For full details, see our code release.

For CIFAR-10, CIFAR-20, STL-10, ImageNet-10 and ImageNet-Dogs experiments, we used LARS optimizer (You et al., 2017), with base batch size 256, for other experiments we used Adam. We note that stage 2 and 3 in the 3 stage training process is quite sensitive to weight decay, a careful search of this parameter is usually required for good performance.

### B.4 COMPUTATIONAL COST

All experiments involving ResNet-34 requires 8 GPUs, others can be done in 1 GPU.

Our objective doesn't add significant computational burden compared to neural-networks involved. The covariance matrix is computed within a batch. For a batch of latent feature vectors with shape $[B, d]$, where $B$ is batch size and $d$ is latent dimension. We first compute the covariance matrix with shape $[d, d]$, with $\mathcal{O}(B^2 d)$ cost. Then the log determinant of this matrix is calculated, which we assume has $\mathcal{O}(d^3)$ cost. The $\mathcal{O}(B^2)$ scaling with respect to batch size is the same as most contrastive learning method such as SimCLR, which is known to scale to very large batch size.

Table B.1: Hyper-parameters for all experiments. lr: learning rate. wd: weight decay. $\epsilon$: coding error. $d_z$: dimension of feature output. $\lambda$: regularization constant. bs: batch size. epochs (steps): total epochs trained, or total steps trained if the entire dataset is passed at once. S1, S2, S3 denote 3 training stages.

| Model | lr | wd | $\epsilon$ | $d_z$ | $\lambda$ | bs | epochs (steps) |
|---|---|---|---|---|---|---|---|
| Double Spiral | 1e-3 | 1e-6 | 0.01 | 6 | 4000 | 4096 | 30000 |
| Synthetic | 1e-3 | 1e-6 | 0.01 | 12 | 100 | 4096 | 3000 |
| COIL-20 | 1e-3 | 1e-6 | 0.01 | 40 | 20 | 1440 | 2000 |
| COIL-100 | 1e-3 | 1e-6 | 0.001 | 200 | 20 | 7200 | 10000 |
| CIFAR-10 ResNet-18 S1 | 1 | 1e-6 | 0.2 | 128 | 50 | 1024 | 600 |
| CIFAR-10 ResNet-18 S2 | 0.5 | 0.005 | 0.2 | 128 | 50 | 1024 | 100 |
| CIFAR-10 ResNet-18 S3 | 0.003 | 0.005 | 0.2 | 128 | 50 | 1024 | 100 |
| CIFAR-10 ResNet-34 S1 | 1 | 1e-6 | 0.2 | 128 | 50 | 1024 | 1000 |
| CIFAR-10 ResNet-34 S2 | 0.5 | 0.001 | 0.2 | 128 | 0 | 1024 | 100 |
| CIFAR-10 ResNet-34 S3 | 0.003 | 0.001 | 0.2 | 128 | 0 | 1024 | 100 |
| CIFAR-20 ResNet-18 S1 | 1 | 1e-6 | 0.2 | 128 | 50 | 1024 | 600 |
| CIFAR-20 ResNet-18 S2 | 0.5 | 0.001 | 0.2 | 128 | 0 | 1024 | 100 |
| CIFAR-20 ResNet-18 S3 | 0.003 | 0.001 | 0.2 | 128 | 0 | 1024 | 100 |
| CIFAR-20 ResNet-34 S1 | 1 | 1e-6 | 0.2 | 128 | 50 | 1024 | 1000 |
| CIFAR-20 ResNet-34 S2 | 0.5 | 0.002 | 0.2 | 128 | 0 | 1024 | 100 |
| CIFAR-20 ResNet-34 S3 | 0.003 | 0.002 | 0.2 | 128 | 0 | 1024 | 100 |
| STL-10 ResNet-18 S1 | 1 | 1e-6 | 0.2 | 128 | 30 | 1024 | 1000 |
| STL-10 ResNet-18 S2 | 0.5 | 0.002 | 0.2 | 128 | 0 | 1024 | 400 |
| STL-10 ResNet-18 S3 | 0.0005 | 0.002 | 0.2 | 128 | 30 | 1024 | 400 |
| STL-10 ResNet-34 S1 | 1 | 1e-6 | 0.2 | 128 | 30 | 1024 | 2000 |
| STL-10 ResNet-34 S2 | 0.5 | 0.002 | 0.2 | 128 | 0 | 1024 | 400 |
| STL-10 ResNet-34 S3 | 0.0005 | 0.002 | 0.2 | 128 | 30 | 1024 | 400 |

