# OpenReview forum: "Neural Manifold Clustering and Embedding"
_ICLR.cc/2022/Conference — ICLR 2022 Submitted_

### Official Review · Reviewer_dgFk · 2021-10-26

**Correctness:** 3
**Technical Novelty And Significance:** 3
**Empirical Novelty And Significance:** 3
**Recommendation:** 6
**Confidence:** 4

**Main Review:**

Fundamentally the work is based on the existing Maximum Coding Rate Reduction (MCR2) objective (Yu et al., 2020). Through their analysis, to achieve the goal of manifold learning in unsupervised fashion, in principle the authors emphasize the importance and necessity of geometry-awareness constraint to be added upon the MCR2 objective. The constraint is surprisingly implemented by the simple data augmentation for example of adding small noises to the given data. And then apply the intuition that the learnt feature embedding of augmented data points should be similar if they are generated from the same point as the geometric constraint. I think this is a very interesting and useful way to enforce the local structures in the data. The authors show a toy example with their code for the performance of the proposed principle, although no code for reproducing large experiments is provided. It is hard to say what performance could be given the high complexity of data while a simple data-augmentation strategy is used, although the past contrastive learning objective works well in practice. Of course, there is much theoretical gaurantee for the introduced constraint strategy.

The paper can be further improved by increasing readability.   For example I dont think Definition 1 is a good way to introduce Manifold Clustering and Embedding task.  A definition should be used for the rigorous, concise and accurate "mathematical" concepts. For example, the "perpendicular" may be written as a definition.

Can you give some discussion on the dimensions d1,d2,...,dn?  What is their role in the algorithm?  Should they be known or can be learned from the algorithm?  Otherwise, there is no need to present them.   It seems the algorithm has nothing to do with them.

You may add how cov(Z) is handled in the algorithm?  What is the algorithm complexity given that the model parameters are inside latent variable Z?

**Summary Of The Paper:**

This paper enhances the performance of Maximum Coding Rate Reduction (MCR2) objective in subspace feature learning by incorporating a strategy of adding a manifold learning inducing constraint, aiming at a novel method for general purpose manifold clustering. They claim that the new model significantly outperforms autoencoder-based deep subspace clustering and achieves
state-of-the-art performance on several important benchmarks.

**Summary Of The Review:**

The proposed method builds upon the existing existing Maximum Coding Rate Reduction (MCR2) objective with a data augmentation as a constraint to enforce manifold learning. It seems that the performance is satisfactory comparing with other state-of-the-art methods.

---

> ### Author Response · Authors · 2021-11-22
> **Reply to Reviewer dgFk**
>
> We want to thank the reviewer for the detailed review!
>
> Based on your suggestion, we will change the ‘Definition’ to ‘Task’, which might be a better word to use. Further, we will revise the paper accordingly to make it easier to read.
>
> Your comment about the dimension of the ground truth manifold is correct. At this point, the proposed algorithm has no way of learning the dimension of each sub-manifold. In fact, we observe that it tends to learn a set of subspaces in the embedding space with equal dimensionality.
>
> Regarding your comment on computing costs: During training, cov(Z) is calculated for each batch: for a batch of latent vector z with shape [B, d], where B is batchsize and d is latent dimension. We first compute the covariance matrix with shape [d,d] by matrix multiplication, with $O(B^2)$ cost, and then compute logdet of the covariance matrix, which we assume has $O(d^3)$ cost. The backward pass supposedly has the same complexity. In our experience, this objective doesn’t incur a significant computational burden compared to the neural network.

---

### Official Review · Reviewer_zSx8 · 2021-11-02

**Correctness:** 3
**Technical Novelty And Significance:** 2
**Empirical Novelty And Significance:** 2
**Recommendation:** 6
**Confidence:** 3

**Main Review:**

# Paper strengths #
The paper suggests a loss function for learning manifold clustering and embedding by combining two ideas from the literature. First, building on a technical paper introducing the “maximal coding-rate reduction” principle which suggests that to learn a subspace-structured representation, one needs to maximize the difference between the coding rate of all clusters (pooled) and the sum of coding rates for each cluster. Second, building on the insight (shared with related works detailed in the “Clustering with Data Augmentation” section) that data-augmentation provides an important signal for manifold embedding, as preservation of class membership across augmented samples replaces preservation of local geometry from classic manifold learning. While both ideas are not new per se, it is interesting that the current paper outperforms both lines of work by using a combined loss function (named NMCE, as the paper initials), suggesting a fruitful synergy between those two ideas.
The optimization target allows using any deep network as a manifold embedding function in either an unsupervised or supervised manner. The unsupervised method (using a simpler loss function, named TCE) nicely resembles recent self-supervised methods, while the supervised method is used as fine-tuning of the unsupervised method and seems to work well.

# Paper weaknesses #
The paper structure does not follow the logical structure presented in my summary: while the combined loss function is well described and justified, the two algorithms are presented as an afterthought. It is not clear why the unsupervised method is needed, while the supervised method is not presented at all. Restate the paper contribution to make the distinction between the loss function, the unsupervised, and the supervised methods.
Present the unsupervised method by itself (rather than as an empirical note in section 3.6) and elaborate on the comparison to self-supervised methods (currently in “additional results” appendix). Explain why the unsupervised method is needed (my guess is this is due to zeroing out of the NMCE objective if class assignments are randomly initialized)
The supervised method is not adequately described, which seems like a major omission. If I correctly filled in the blanks, a training set was used for minimizing the NMCE objective function using the ground-truth class assignment (i.e. only the embedding is learned), then clustering success is measured on a test-set with some unknown initialization of class assignment. This initialization may be important for the results if it is not random (e.g., by using the training set and finding the nearest neighbor).
No motivation is given for the role of such work in the ML landscape. The paper starts with a technical assumption (“a union of low-dimensional nonlinear manifolds”) rather than from an intuitive statement that (1) each sample comes from some class; (2) classes representation has a meaningful structure, conceptualized as a low-dimensional nonlinear geometry; (3) it is natural to ask if class-membership and a low-dimensional geometry can be learned simultaneously.
Important principles are hidden in technical definitions and the authors should aim to provide more intuition. Most notable are the mysterious references to MCR2. If “Gaussian coding rate” is a quantification of variance, “Maximum Coding Rate Reduction (MCR2)” supports maximizing the total variance while minimizing  “variance within clusters”. Similarly the “constraint functional D(f)” is presented in section 3.2 as a “completely generic” method and only in section 3.5 do we understand how to implement it by enforcing similar clustering of data-augmented samples. I find the generic presentation unhelpful while the use of data-augmentation to uncover manifold structure central to the paper.
The main method presented in the paper does not seem to work well on synthetic data sets, and the authors do not share the details. Summarize the difference between EnSC and the full loss functions and explain why “For simple tasks, the features can be directly clustered with standard linear SC techniques such as EnSC”.  Also, provide the (poor) result of the full algorithm.

## Smaller issues ##
 (1) The vogue statement “This work shows that many constraints can be implemented by data augmentation” should be re-stated through the lens of self-supervised learning, e.g., “manifold local structure can be learned by enforcing constant class-assignment on data-augmented samples”

 (2) Highlight the difference from the related work in the “Clustering with Data Augmentation” section and state in advance that current work outperforms them.

 (3) The statement “we need augmentations that perturbs style information (...) but preserves content information, so that the clustering will be based on content but not style” seems weird as this is almost the definition of data augmentation.

 (4) The statement “This indicates that NMCE can truly leverage the non-linear processing capability of deep networks” seems false in the context of section 4.1, where the TCR objective is used and not the full NMCE objective.

 (5) In section 4.2.1 the role of stage 2 (“Reinitialize the last linear projection layer and add a cluster assignment layer … freeze parameters in the backbone network and train the two new layers with full NMCE objective”) is not clear. If I understand correctly, you don’t show results of stage 2 without stage 3 and you don’t justify that stage 2 is needed. You should provide some evidence stage 2 is helpful or if not skip it for the sake of simplicity.

 (6) I don’t understand the discussion which makes most of section 4.2.1, about using representations at different levels of the embedding network, and with or without averaging. It seems the authors had several options and made their choice, so this section and table 2 can go to the “additional results” appendix to justify the choice.


**Summary Of The Paper:**

The paper “Neural Manifold Clustering and Embedding” proposes methods for utilizing a deep neural network for simultaneously learning the clustering of samples and a non-linear embedding into a low-dimensional “latent space” where each cluster representation is simple.
The main contributions of this paper are:

 (1) Proposing a novel loss function that combines the “maximal coding-rate reduction” principle and self-supervised view of how to use data-augmentation in clustering;
 (2) Proposing an unsupervised method for solving the problem from scratch using a deep neural network as embedding operation, with parameters learned to maximize “coding-rate reduction” while preserving class membership across augmented samples;
 (3) Proposing a supervised method for fine-tuning the results of the above method by minimizing the “coding-rate reduction” of each class while maximizing the overhaul  “coding-rate reduction” and preserving class membership across augmented samples.


**Summary Of The Review:**

The paper combines two previous lines of work into a novel loss function which is then used in both supervised and unsupervised manners and achieves state-of-the-art results. It is currently below the acceptance threshold because the presented algorithms’ description and justification are lacking.

---

> ### Author Response · Authors · 2021-11-22
> **Reply to Reviewer zSx8**
>
> We want to thank the reviewer for the detailed review and various suggestions.
>
> First, we want to clarify a potential point of confusion. In your review, a “supervised algorithm” is mentioned, and the feature is learned with “ground truth class assignment”. We want to emphasize that we only deal with unsupervised clustering in this paper. The class label is used only to evaluate the resulting clustering performance and nothing else. I think your interpretation best describes the original MCR$^2$ algorithm, which uses ground truth labels as the subspace assignment and learns a subspace-structured feature space. In our method, we optimize the cluster assignment and feature embedding simultaneously, which does not use the ground truth label.
>
> To your comment regarding the constraint function: It is true that we only studied constraints implemented via data-augmentation in our paper. However,  the definition of the constraint function allows us to cover essentially any clustering algorithm, allowing neural networks to be applicable. For example, we can also consider building a constraint function that computes the average local curvature of points assigned to the same cluster, which would have the effect of constraining the learned cluster to be smooth.
>
> We are unsure why you commented that “The main method presented in the paper does not seem to work well on synthetic data sets”. In the main text, we used the double spiral synthetic dataset to demonstrate the main idea, and the clustering accuracy is clearly close to 100%. In the appendix, we further showed the performance of our proposed method on a synthetic dataset of manifolds generated with randomly initialized neural networks. The resulting clustering accuracy is still 100%. In the second toy example, the learned feature’s behavior is also consistent with one's expectations. Where the feature is near perpendicular between clusters, and not perpendicular within a cluster.  Your impression could have been caused by the “Not-identifiable” result in Table A.1 in the appendix. This result represents the case where the manifolds are intersecting, thus are not identifiable via only locality constraints. The clustering algorithm does not reach high accuracy in this case, which is also expected.
>
> Regarding your comment about our statement in the paper: “This indicates that NMCE can truly leverage the non-linear processing capability of deep networks”: Sorry for the potential confusion, but here we are actually comparing our method to other subspace clustering algorithms that use neural networks. In principle, using neural networks enables the clustering of non-linear manifolds. However, it has been shown that these techniques outperform linear subspace clustering mostly because of a post-processing step on the adjacency matrix, instead of the non-linear processing capability of neural networks. See [1] and also reference in our paper.
>
> To your issue (5), the role of the second stage is to cluster the features learned during stage 1. And stage 3 further fine-tunes the entire network with the full NMCE objective. We show the effect of stage 2 and stage 3 in appendix table A.2. Stage 2 already gives most of the clustering performance, while stage 3 further improves it a little bit.
>
> Section 4.2.1 discusses the performance of various feature spaces learned with this algorithm. We showed that under some settings, the low-dimensional features in the latent space outperform the features after the pooling layer. Our finding is contrary to the conventional wisdom of using the pooling layer feature to perform classification and throwing away the embedding feature. We believe this result is notable and interesting to the self-supervised learning community, therefore we presented it in the main text.
>
> [1] Benjamin D Haeffele,  Chong You,  and Ren ́e Vidal.   A critique of self-expressive deep subspace clustering.arXiv preprint arXiv:2010.03697, 2020

---

> > ### Comment · Reviewer_zSx8 · 2021-11-24
> > **Reply to authors**
> >
> > Thank you for your clarification! It seems I made two mistakes in understanding your paper:
> >   1. Misunderstanding that your method is fully unsupervised and the ground-truth are not used in fine-tuning;
> >   2. Believing only stage (1) is used for the simpler data sets (due to the confusion in the algorithm name in the results table).
> >
> > Given that, I believe the paper's results justify publication and I would like to change my score from 5 to 6.
> >
> > Having said that, I can't escape the feeling that my mistakes are not completely my failure; the paper is not well written and my comments regarding the style of the presentations were not addressed:
> >  * No motivation is given for the role of such work in the ML landscape.
> >  * The attempt to write a "completely generic" definition of clustering-related constraints D(f) is confusing and hides one of the contributions of the paper, namely the use of data augmentation to uncover manifold structure.
> >  * No intuition is given for MCR2 (I suggested "maximizing the total variance while minimizing variance within clusters")
> >
> > For me it's a shame the authors' responses came after the end of the period of allowed changes and that the authors did not try to improve their presentation style within that period.

---

> > > ### Author Response · Authors · 2021-11-25
> > > **Reply to Reviewer's reply**
> > >
> > > Thanks for the reply and it's great to see that your confusion is resolved.
> > >
> > > We appreciate your feedback on the writing of our paper, and we are sorry that we missed your suggestions of edits. Adding an intuitive explanation to MCR$^2$ objective as you stated is indeed a very helpful way to introduce it! We will for sure add that to the paper for the camera-ready version (if we have the opportunity to submit one), or future publication of this paper.
> > >
> > > We believe that the position of our work in the ML field is that we provide a unified view of non-linear subspace clustering and self-supervised learning (clustering), by empirically showing that in the case where the constraint function is implemented by data-augmentation, we can perform both with essentially the same algorithm. The purpose of the generic constraint functional $D(f)$ is to formalize this unified view. Although admittedly it's slightly confusing, we feel that without it, the paper will be too focused on a specific technique and be less well-motivated.

---

### Official Review · Reviewer_aRVQ · 2021-11-02

**Correctness:** 3
**Technical Novelty And Significance:** 2
**Empirical Novelty And Significance:** 3
**Recommendation:** 5
**Confidence:** 3

**Main Review:**

The paper provides an intuitive extension of the MCR^2 embedding approach by regularizing it with a "concordant data augmentation embedding" constraint. The authors reason that such a constraint is needed to "[restrict] the large flexibility of neural networks".

However, the paper is imprecise in many of the statements made. A reference to "additional geometric constraints that make the manifold clusters identifiable" essentially reduces to proximity of samples and their noisy versions in the manifold embedding. $L_{clst}$ is described as "some objective function that will force $f$ to cluster the dataset" but again there is no practical example of such a function. The concept of a constraint functional $D(f)$ is also never specified for a practical setting. In the appendix the authors say "the performance of [MCR^2] is rather poor, which is expected based on our understanding" - this does not illuminate a reason for the reader or make it clear what "CTRL" brings in.

It is not clear why only a partial subset of the code that implements the experiments is provided with thee rest to be held for release "upon publication". Also there are no descriptions of the computation cost of the different approaches compared, including the proposed one.

The fact that the augmentation/regularization used in the numerical experiments is tuned by hand raises further questions about the robustness of the proposed approach. The authors should provide some discussion on this aspect of the implementation - this can be done in the appendix if needed.

The training procedures used in Section 4.2 appear to be ad-hoc - can the authors comment on their reasoning? Additionally, some exclusions in the numerical comparison seem arbitrary as well - if a method in the literature addressed the same problem considered here, why not compare against it? If there are differences in approaches they can still be mentioned to provide a contrast.

Why is there no numerical comparison in the experiments shown on the appendix?

There is a repeated typo (e.g., in Fig. 2 and Fig. A.3) captions "principle components" -> "principal components".

**Summary Of The Paper:**

The paper considers the problem of clustering a dataset corresponding to a sampling of a union of nonlinear manifolds where each cluster shall correspond to an individual manifold. The proposed method leverages previous results on subspace clustering via neural networks and a penalty function based on concordant classification of samples and their augmented versions.

**Summary Of The Review:**

While the extension of a supervised multi-manifold embedding to the manifold clustering problem is intuitive and interesting, there are several imprecise concepts underlying its description, and the carefully crafted implementation muddles whether the proposed approach can have wider applicability.

---

> ### Author Response · Authors · 2021-11-22
> **Reply to Reviewer aRVQ**
>
> Thanks for your detailed review and corrections.
>
> To your point about the constraint functional: We agree that such a functional is idealized, and so far the only implementation we are considering is data augmentation, which can realize a simple locality constraint or a more complex constraint that allows clustering of images. However, one can easily imagine another type of practical constraint functional implementation. For example, a constraint that estimates the local curvature of found clusters, which would correspond to a smoothness constraint. The constraint functional is an idealized tool that allows us to understand manifold clustering and self-supervised learning in a unified conceptual framework, which we believe is our main contribution.
>
> To your concern about tuning augmentation by hand: As we discussed, the constraint function reflects the user's knowledge about the target space where the algorithm is applied. Therefore, some level of manual design is inevitable. For example, the noise scale in the locality constraint is chosen by visualizing the distribution, the data augmentation for self-supervised learning of image datasets is adopted from SimCLR, where the augmentation is found empirically by optimizing the accuracy on ImageNet.
>
> We have also added an ablation study of the effect of the lambda parameter (that balances the constraint term with the coding rate term) in self-supervised learning with the TCR objective. The result is listed in the Appendix of the updated draft. It seems that our algorithm works reasonably well for a range of parameter settings.
>
> |$\lambda$| 20 | 30 | 40 | 50 | 60 | 70 |
> |--|--|--|--|--|--|--|
> |Proj SVM acc|0.899|0.902|0.903|0.902|0.903|0.903|
> |Proj kNN acc|0.890|0.895|0.894|0.896|0.895|0.897|
>
> We omitted [1] and [2] in our clustering performance comparison because [1] is a post-processing step that can be applied to any clustering algorithm, rather than a clustering algorithm by itself. [2] uses a special network architecture, thus, the comparison would not be fair.
>
> [1] Sungwon Park,  Sungwon Han,  Sundong Kim,  Danu Kim,  Sungkyu Park,  Seunghoon Hong,  andMeeyoung Cha.  Improving unsupervised image clustering with robust learning.  In Proceedings of the IEEE/CVF Conference on Computer Vision and Pattern Recognition, pp. 12278–12287,2021.
>
> [2] Vitaliy  Kinakh,  Slava  Voloshynovskiy,  and  Olga  Taran.   Scatsimclr:  self-supervised contrastive learning with pretext task regularization for small-scale datasets.  In2nd Visual Inductive Priorsfor Data-Efficient Deep Learning Workshop, 2021.

---

> > ### Comment · Reviewer_aRVQ · 2021-11-29
> > **Thank you**
> >
> > Thank you for your response. It seems we agree on the need for hand-tuning many aspects of the algorithm, and the dependence of the performance of the algorithm on the value of  $\lambda$ seems appropriate; it remains to be seen how a lower value of $\lambda$ would affect this. It would also be good to hear from the authors about whether any changes would be made to the manuscript to address the points discussed here (e.g., add these explanations in the manuscript?)
> >
> > We still disagree on the explanation for excluding the numerical results for [1-2]. They could be added with the explanations you are providing here.

---

> > > ### Author Response · Authors · 2021-11-30
> > > **Further Discussion**
> > >
> > > Thanks for your reply!
> > >
> > > In the actual experiments, we chose $\lambda = 50$; we believe $20$ - $70$ is a reasonable range to show that the performance is relatively stable w.r.t. this hyperparameter. Based on the trend, we expect the performance to continue to decrease slightly with lower $\lambda$.
> > >
> > > We will add additional explanation to the paper to clarify that the constraint functional is an idealized object that needs to be relaxed to be used in practice.
> > >
> > > As we discussed in the previous response, [1] can be potentially combined with our method. Further, the proposal in [2] is relatively orthogonal to this paper. In the next revision, we can also discuss the numerical results from [1] and [2], explaining that they are not directly comparable to our method.

---

### Official Review · Reviewer_SD19 · 2021-11-03

**Correctness:** 2
**Technical Novelty And Significance:** 2
**Empirical Novelty And Significance:** 2
**Recommendation:** 5
**Confidence:** 4

**Main Review:**

### Strengths & Originality:

The idea of combing "Maximum Coding Rate Reduction" with data argumentation techniques is interesting and seems reasonable.

In the experimental parts, this work have results on a number of real data sets, which can be considerate a fairly enough set up to support the effectiveness of a newly proposed algorithm.

### Weakness:

Overall it seems NMCE is like a relative less significant changes from the work from recent proposed "Maximum Coding Rate Reduction (MCR2) (Yu et al., 2020)" with some data argument techniques, and there is no interesting theoretical analysis behind the proposed NMCE work too, from this paper.

Alternative methods: in the experimental results & section 4, there are a number works are cited and compared to NMCE but still lack some important references. For example in Table 1, for COIL data set, results from linear methods like SSC and some deep clustering methods are listed but there is a whole field of nonlinear manifold clustering is not mentioned here. Reference 1 and 2 listed below are two methods in the field of nonlinear manifold clustering and some in-depth discussion or comparison to them are quite helpful for reviewer to evaluate the contribution from NMCE. Indeed, there is one survey paper about nonlinear manifold clustering is cited in section 2 but this is not enough.

To the toy example, double spiral, it seems NMCE can get 100% accuracy as shown in Appendix A. However, only one 100% number is kind of less clear as we can imagine that if keep increasing the scale of Gaussian noise in the data argumentation step, we should except this 100% to go down. Or maybe we can say the key is how to define "a small amount of noise" as stated in the paper, and should be nice to see the performance from alternative methods on toy examples too, to gave people more insights, i.e., multi-linear subspace clustering methods clearly not work well for this toy example as we know.

### Reference:
1. Souvenir, R., & Pless, R, Manifold clustering. The10th International Conference on Computer Vision (ICCV 2005).
2. Dian Gong, Xuemei Zhao, Gérard G. Medioni, Robust Multiple Manifold Structure Learning. ICML 2012.

**Summary Of The Paper:**

This work proposed a general manifold clustering algorithm called Neural Manifold Clustering and Embedding (NMCE), which utilize Maximum Coding Rate Reduction (MCR2 ) as the objective function and data augmentation to enforce constrains.

In the implementation stage, given that even the toy experiment is difficult to optimize with the full NMCE objective, a multistage training procedure is applied with the first stage actually trying to optimize the Total Coding Rate (TCR), which is another kind of self-supervised learning objective claimed in this paper.

On synthetic and real-world datasets COIL20, COIL100, CIFAR-10, CIFAR-20 and STL-10, NMCE achieved comparable and sometimes better results, compared to baseline methods or some alternative manifold clustering methods listed in the paper.

**Summary Of The Review:**

Overall I feel this is a reasonable submission but seems not good enough for ICLR publication, so I gave " marginally below the acceptance threshold ".

---

> ### Author Response · Authors · 2021-11-22
> **Reply to Reviewer SD19**
>
> Thanks for the detailed review and suggested references. We have added the suggested references with a brief discussion in our related works section. Indeed these papers are highly relevant to our work.
>
> Reference 1 in your review uses local distance and an E-M type algorithm to perform manifold clustering. Reference 2 additionally uses a curvature constraint to make the case of intersecting manifolds identifiable. Our constraint function generalizes both of those concepts so in principle they can also be implemented with Neural Networks. However, we are unable to find a ready-to-use implementation of the methods in those 2 papers, so it is difficult to compare performance on the COIL dataset with them. We also didn’t find any previous work that reported the performance of those baselines. If there is a suggested implementation or we could find one later, we would add more comparisons.
>
> To your comment that our work is a less significant change from MCR$^2$: We would like to emphasize that our work is not a straightforward combination of MCR$^2$ with data augmentation. The original MCR$^2$ work also used data-augmentation when performing self-supervised learning. In that paper, augmented samples from one data point are treated as a subspace. Because the MCR$^2$ objective also maximizes the dimension of each learned subspace, the augmentation information is partially retained in the learned feature space. Our new interpretation of non-linear subspace clustering motivated a different use of data-augmentation with MCR$^2$. In the new interpretation, MCR$^2$ is used as a subspace feature learning method. We note that this is not the only objective for feature learning that can work. In fact, in some of our experiments, using total coding rate (TCR) suffices to learn a subspace-structured representation, although the reason is not entirely clear to us. In the future, better subspace feature learning algorithms could be proposed but the principle that allows non-linear manifold clustering to be achieved will be the same.
>
> On another note, our alternative use of data augmentation significantly improved the self-supervised performance of the MCR$^2$ objective, from 68.4% to 83% accuracy on CIFAR-10 with ResNet-18, for example.
>
> Your interpretation of the effect of the noise scale is correct. The noise scale for the toy dataset needs to be empirically chosen, or chosen based on prior knowledge about the dataset. Too much noise will cause “shortcuts” between different manifolds, and too little noise makes the constraint too weak to identify the manifold. We did not run our toy example on classical manifold clustering methods, but intuitively, we can expect methods like that in reference 1 of your review to also reach 100% accuracy, as it is also a locality-based manifold clustering algorithm. The point of the toy example is not to compare performance, but to show that a unified interpretation and algorithm based on neural networks work in a wide range of settings. The range goes from problems easily solvable by classical manifold clustering algorithms to the challenging problem of the clustering of high-dimensional natural images. That, in our opinion, is not trivially achieved.

---

> > ### Comment · Reviewer_SD19 · 2021-11-30
> > **Reply to Authors**
> >
> > Thank you for providing additional colors & revisions to help us in this discussion process.
> >
> > 1. Reference: I agree it seems can not find ready-to-use implementation of the methods in those listed 2 papers in my comments, in a way that these 2 references are more like 2 selected paper from the "field of nonlinear manifold clustering", some in-depth discussion or comparison (when possible) to this field is quite helpful to see.
> >
> > 2. Maximum Coding Rate Reduction: indeed interesting to see the direction of utilizing MCR2 as the subspace feature learning method, given this, it still seems that majority of the methodology side contribution belongs to the original MCR2 work.
> >
> > 3. Noise scale: thanks for confirming , good to know & the remaining interesting and kind of unclear part for me is this "empirically chosen, or chosen based on prior knowledge".

---

> > > ### Author Response · Authors · 2021-11-30
> > > **Further Clarification**
> > >
> > > Thanks for your reply!
> > >
> > > To your remaining question about noise scale being  "empirically chosen, or chosen based on prior knowledge." We mean that it is chosen by optimizing the clustering performance or simply by looking at the geometry of the data space and guessing an appropriate scale, as it was done in the double spiral experiment. We can further clarify this in the next revision.

---

### Official Review · Reviewer_HTFw · 2021-11-04

**Correctness:** 3
**Technical Novelty And Significance:** 2
**Empirical Novelty And Significance:** 2
**Recommendation:** 3
**Confidence:** 4

**Main Review:**

Manifold clustering is a very hard problem. Authors propose a solution for non-linear manifold clustering. The aim of the paper is to learn a representation space in which the manifolds are co-linear. The problem is quite interesting and there are practical applications. However, I have several concerns:
- I believe two highly relevant papers are missing in the paper and these papers are
[1] Clustering-friendly Representation Learning via Instance Discrimination and Feature Decorrelation (ICLR 2021)
[2] Representation Learning for Clustering via Building Consensus (arxiv, May 2021)
I am completely aware these papers don’t consider manifold clustering however they can be used in this setting as they are. I will lay out my concerns by referring to these two missing papers.
Authors rightfully highlights the importance of assumptions and constraints in their paper. One of the proposed constrained is orthogonality (Section 3.1). Both [a] and [b] utilizes orthogonality. [1] directly use originality in their loss and [2] uses random projections.
Authors are trying to learn one representation space for different manifolds. Similar objective can be achieved by consensus loss of [2].

- The objective given in eq 3 is interesting. In my understanding, objective wants to condense each manifold as much as possible (summation term) and all manifolds should cover the largest available space (first term). I think these conditions will be satisfied if all manifolds are equidistant from each other. This can be desired property however how realistic it is? For example, let’s consider the following classes {car, bus, motorbike, bicycle, lion, mountain lion, whale, lion}. Why should each manifold be equidistant from each other? One would like to see semantic structure in the learned representation. Would you please explain how will eq 3 type loss will handle this issue?

-	The proposed loss depends on logdet (please see Eq 2). As far as I know, to calculate such a quantity, one needs to calculate the eigen values of the cov(Z). I am assuming that the authors are calculating this quantity for each batch. Since eigen values needs to get calculated by another algorithm, e.g. Gaussian elimination,
o	does this mean the proposed loss cannot generalize to large batch sizes? Some of the self-supervised learning methods are trained by using 8192 batch size.
o	how can one compute logdet in multiple gpus? Does one need to do this computation on one gpu?

Experimental results:
-	Would you please compare your CIFAR-10, CIFAR-20 and STL-10 results with [1] and [2]? Please note that the results shown in [1] and [2] may be using different architectures. As far as I remember [1] use Resnet-34 and [2] use Resnet-18.
-	Both [1] and [2] are end-to-end training methods. Especially [2] supplies better results for Resnet-18 setting. Would you please elaborate the advantage of multi-stage training given that end-to-end are suppling similar results?
-	Would you please extend your empirical studt to ImageNet-10 and ImageNet-Dogs.
-	Would you please supply some ablation studies e.g. impact of lambda parameter?

I am looking forward to author responses and I am willing to change my assesment/score if the reponse(s) is(are) satisfactory.

**Summary Of The Paper:**

Authors developed a method and training procedure for manifold clustering problems. The proposed solution inspired from information theoretic methods namely maximum rate reduction. Authors also support their claims with empirical results.

**Summary Of The Review:**

Although authors proposed an interesting solution to a very hard problem, I believe proposed method and experimental work needs to be improved especially taking in to account missing references and supplying a detailed discussion.

---

> ### Author Response · Authors · 2021-11-22
> **Reply to Reviewer HTFw**
>
> Thanks for your detailed review and pointing to missing references. Indeed, the two papers are very relevant to our work, and we have updated our draft to discuss them in the Related Works section, as well as comparing performance with them. The following is our understanding of the relationship between our work and the references you mentioned:
>
> Reference [1] in your review studies the combination of instance discrimination with feature decorrelation and orthogonization for self-supervised contrastive learning. The first stage in our multi-stage training procedure, the TCR objective, turns out to be equivalent to flattening the singular values of the covariance matrix, which is equivalent to feature decorrelation.
>
> Reference [2] in your review proposes clustering by enforcing consensus consistency, which refers to different clustering results after random projection. In this paper they introduced two other concepts of consistency, namely exemplar consistency and population consistency. Those two concepts are closely related to the constraint we proposed. The idealized concept of the constraint function in our paper essentially enforces population consistency on the learned clustering function.
>
> Your interpretation of Eq3 in our paper is correct. The subspace feature learning loss in Eq3 will make the learned subspace clusters orthogonal to each other, no matter what the semantics of those clusters are. Our interpretation is that this loss function does not encode the desired semantics. Rather, the semantics is enforced by the constraint function and the dataset. For example, when clustering Imagenet, different kinds of dogs will likely reside in the same subspace, as they are similar to each other compared to images from other classes. If one optimizes this objective on the Imagenet-dogs dataset, different breeds of dogs will be pushed into  approximately orthogonal subspaces, as there isn’t any other class to contrast with dogs.
>
> Regarding the computation cost of the log determinant of the covariance matrix: Consider a batch of latent vector z with shape [B, d], where B is batchsize and d is latent dimension. We first compute the covariance matrix with shape [d,d], with $O(B^2)$ cost, and then compute the logdet of the covariance matrix, which we assume has $O(d^3)$ cost. The $O(B^2)$ scaling with respect to batchsize is the same as other contrastive methods like SimCLR, which also compute dot product of all pairs of samples in a batch. Therefore, we do not expect significant scaling issues to larger batchsize. We have added a discussion of computation cost to our updated draft.
>
> Regarding the use of multi-stage training: As discussed in the paper, we have tried training end-to-end from scratch without success. Our interpretation is the following. At initialization, the cluster assignment is not meaningful, which causes the sum and the total coding rate term in Eq3 to cancel, providing no meaningful gradient. It is possible to initialize the training by first down-scaling the sum term, and gradually scale it back at a later stage. We feel it’s simpler to just down-scale to 0 in the first stage, which produces the TCR objective.
>
> We have added the  experiments you suggested on ImageNet-10 and ImageNet-Dogs using ResNet-18. Specifically, our method achieved 90.6 accuracy on imagenet-10, and 39.9 accuracy on imagnet-dogs, which is on par with techniques like CC [1]. The performance metrics are also listed in the updated main text.
>
> Further, we have also added an experiment that compares CIFAR-10 self-supervised accuracy under different lambda parameters in the Appendix of the updated manuscript. The performance is reasonable over a fairly large range of lambda values. Other self-supervised algorithms also have parameters that need to be tuned, for example, the temperature parameter in SimCLR is known to affect the result significantly. We also copy the table from the appendix in the following.
>
> |$\lambda$| 20 | 30 | 40 | 50 | 60 | 70 |
> |--|--|--|--|--|--|--|
> |Proj SVM acc|0.899|0.902|0.903|0.902|0.903|0.903|
> |Proj kNN acc|0.890|0.895|0.894|0.896|0.895|0.897|
>
> [1] Yunfan Li, Peng Hu, Zitao Liu, Dezhong Peng, Joey Tianyi Zhou, and Xi Peng. Contrastive cluster-ing. In 2021 AAAI Conference on Artificial Intelligence (AAAI), 2021.

---

> ### Author Response · Authors · 2021-12-09
> **Reminder for comment**
>
> Hi,
>
> In case you missed our response message. We have added various experiments as requested, and reponded to many of your questions. Please kindly let us know if our response is satisfactory.
>
> Many thanks.

---

### Author Response · Authors · 2021-11-22
**General response**

We thank reviewers for taking the time to provide many insightful comments that helped us to further improve this paper! We believe the main contribution of this paper is to establish a connection between non-linear subspace clustering and several important self-supervised learning methods. Empirically, we show that data augmentation can realize constraints that enable neural networks based non-linear subspace clustering from toy tasks all the way to image clustering tasks.

---

### Decision · Program_Chairs · 2022-01-20

**Decision:**

Reject

**Comment:**

The paper received a majority voting of rejection, although the author response successfully convinced one reviewer to increase his/her score from 5 to 6. I have read all the materials of this paper including manuscript, appendix, comments and response. Based on collected information from all reviewers and my personal judgement, I can make the recommendation on this paper, *rejection*. Here are the comments that I summarized, which include my opinion and evidence.

**Presentation**

The presentation of this paper needs huge efforts to further improve. Several reviewers and I suffered from difficulties to understand the motivation and challenges of this paper. It seems that Section 3.5 is the novelty part of this paper, but I failed to catch their points.

**Contribution**

Two contributions points were claimed in this paper. (1) The combination of data augmentation and MCR$^2$. Without knowing the challenges in this paper, it is difficult to evaluate this point. Based on my current understanding (The presentation heavily affects my understanding), this point is very incremental. (2) The proposed method achieved state-of-the-art performance. This point is problematic. I will explain below.

**Related Work**

The authors failed to notice a huge body of manifold learning work and contrastive clustering work. Some state-of-the-art methods are not included for comparisons.

**Experimental Evaluation**

(1) Lack of state-of-the-art methods; (2) No standard deviation; (3) The experimental results are incomplete; and (4) It seems that the proposed method only achieved high performance on CIFAR-10 and CIFAR-20. I am not the person who requests the authors achieve the best performance on all the datasets. Everyone knows no algorithm always wins. But the authors should provide some analyses on the inferior performance for better understanding the model.

No objection from reviewers was raised to again this recommendation.